

Intercomparison of Open-Path Trace Gas Measurements with Two Dual Frequency Comb Spectrometers
Eleanor M. Waxman[1], Kevin C. Cossel[1], Gar-Wing Truong[1], Fabrizio R. Giorgetta[1], William C. Swann,[1]
Sean Coburn[2], Robert J. Wright[2], Gregory B. Rieker[2], Ian Coddington,[1] and Nathan R. Newbury[1]
[1]Physical Measurement Laboratory, National Institute of Standards and Technology, 325 Broadway,
Boulder, CO 80305
[2]Precision Laser Diagnostics Laboratory, University of Colorado Boulder, Boulder, CO 80309
Correspondence to: Eleanor Waxman
(eleanor.waxman@nist.gov)
Abstract
We present the first quantitative intercomparison between two open-path dual comb spectroscopy
(DCS) instruments which were operated across adjacent 2-km open-air paths over a two-week period.
We used DCS to measure the atmospheric absorption spectrum in the near infrared from 6021 to 6388
$cm^{-1}$ (1565 to 1661 nm), corresponding to a 367 $cm^{-1}$ bandwidth, at 0.0067 $cm^{-1}$ sample spacing. The
measured absorption spectra agree with each other to within $5\times10^{-4}$ without any external calibration of
either instrument. The absorption spectra are fit to retrieve concentrations for carbon dioxide ($CO_2$),
methane ($CH_4$), water ($H_2O$), and deuterated water (HDO). The retrieved dry mole fractions agree to
0.14% (0.57 ppm) for $CO_2$, 0.35% (7 ppb) for $CH_4$, and 0.40% (36 ppm) for $H_2O$ over the two-week
measurement campaign, which included 23 °C outdoor temperature variations and periods of strong
atmospheric turbulence. This agreement is at least an order of magnitude better than conventional
active-source open-path instrument intercomparisons and is particularly relevant to future regional flux
measurements as it allows accurate comparisons of open-path DCS data across locations and time. We
additionally compare the open-path DCS retrievals to a WMO-calibrated cavity ringdown point sensor
located along the path with good agreement. Short-term and long-term differences between the two
systems are attributed, respectively, to spatial sampling discrepancies and to inaccuracies in the current
spectral database used to fit the DCS data. Finally, the two-week measurement campaign yields diurnal
cycles of $CO_2$ and $CH_4$ that are consistent with the presence of local sources of $CO_2$ and absence of local
sources of $CH_4$.


1. Introduction

Quantitative determination of greenhouse gas fluxes over a variety of temporal and spatial

scales is necessary for characterizing source strength and intermittency and for future emissions
monitoring, reporting, and verification. To this end, techniques exist to measure greenhouse gas
concentrations on a variety of length scales, each of which has advantages and disadvantages. Point
sensors provide valuable information about local sources, but their use for continuous regional
measurements on sampling towers is complicated by local wind patterns, local sources, and mixing
within the planetary boundary layer (PBL), especially at night (Lauvaux et al., 2008; Ciais et al., 2010;
Lauvaux et al., 2012). Similarly, total-column measurements are particularly useful for sub-continental to
global scale measurements; however they are sensitive to atmospheric transport errors within the PBL
(Lauvaux and Davis, 2014), are affected by clouds and aerosols, are primarily limited to daytime
measurements, and lack either the revisit rates or mobility for regional flux measurements. Horizontal
integrated path measurements are complementary to point sensors and satellites: they cover spatial
scales from 1-10s of kilometers and provide measurements on the second to minute time scales with
portable instruments and are thus appropriate for regional studies. Active-source open-path sensors
such as open-path Fourier Transform Infrared spectroscopy (FTIR), differential optical absorption



spectroscopy (DOAS), differential LIDAR (DIAL), or tunable diode laser absorption spectroscopy (TDLAS)
are often used for these measurements and can retrieve path-averaged concentrations but typically
with 10% or greater uncertainties (EPA Handbook, and references therein). Recently, open-path dual-
comb spectroscopy (DCS) has emerged as a new technique that could potentially provide precise,
accurate continuous regional measurements of the mole fractions of $CO_2$, $CH_4$, $H_2O$, and HDO over
kilometer-scale open paths (Rieker et al., 2014), thereby providing a new open-path sensing capability
that falls between point sensing and total-column measurements.
Here we demonstrate that open-path DCS can indeed yield dry mole fractions over open-air
paths with a high level of intercomparibility, over long periods of time, and with sufficient precision to
track variations in the ambient levels from local sources and sinks.  Two completely independent open-
path DCS instruments are operated over neighboring open-air paths during a two-week measurement
campaign.  Although both DCS instruments use fully stabilized frequency combs, they are portable
(Truong et al., 2016) and are operated nearly continuously during both day and night through laboratory
temperature variations from 17 to 25°C, strong atmospheric turbulence, and outdoor air temperature
variations from 4.6 to 28.9°C. The retrieved dry mole fractions for the two DCS instruments agree to
better than 0.57 ppm[1] (0.14 %) for $CO_2$ and 7.0 ppb (0.37 %) for $CH_4$. This agreement is achieved without
any "bias correction" or calibration of either instrument for absolute wavelength or for absolute
concentration. Instead, it is a direct consequence of the negligible instrument lineshape and precise
frequency calibration of the DCS instruments, which leads to measured atmospheric absorption spectra
that are identical to below $10^{-3}$ (limited by the instrument noise level). The measured path-averaged $CO_2$
precision over a 2-km path is 0.90 ppm in 30 seconds, improving to 0.24 ppm in 5 minutes.  For $CH_4$, the
precision is 9.6 ppb in 30 seconds, improving to 2.1 ppb in 5 minutes. We also compare the DCS
retrievals to a WMO-calibrated, cavity ringdown point sensor located near the path. The agreement is
within 3.4 ppm and 17 ppb for $CO_2$ and $CH_4$ respectively, limited by differences in the sampling volume
and by the spectral database used to analyze the DCS transmission spectra.
Similar intercomparison measurements between conventional active, open-path sensors are
rare but have shown agreement of typically 1-20% (Thoma et al., 2005; Hak et al., 2005; Smith et al.,
2011; Shao et al., 2013; Conde et al., 2014; Reiche et al., 2014; Thalman et al., 2015).  Here, we find
agreement between two DCS instruments that is an order of magnitude better and is comparable to
that achieved with highly-calibrated, state-of-the-art, solar-looking FTIRs that retrieve vertical column
measurements (Messerschmidt et al., 2011; Frey et al., 2015; Hedelius et al., 2016); however, open-path
DCS does not require instrument-specific calibrations (e.g. of the instrument line shape) and provides a
very different capability by retrieving the dry mole fractions across regional, kilometer-scale paths over
day and night in a mobile platform. Moreover, as the agreement between open-path DCS instruments is
below the level of natural background fluctuations, future measurements can facilitate accurate inverse
modeling to identify sources and sinks of carbon emission over regions. As an initial demonstration, we
discuss the observed diurnal variations from this two-week measurement campaign in the final section
of the paper.
2 Technique
2.1 Overview
DCS is based on a frequency comb laser source, which is a pulsed laser that outputs a spectrum
consisting of evenly-spaced, narrow modes ("comb teeth") underneath a broad spectral envelope
(Cundiff and Ye, 2003; Hall, 2006; Hänsch, 2006).  In DCS, two such frequency combs are used to
measure the atmospheric absorption on a comb-tooth-by-tooth basis across broad bandwidths

---

[1] In this work we use dry mole fraction.  ppm is defined as micromoles of $CO_2$ per mole of dry air and ppb is defined as nanomoles of $CH_4$ per mole of dry air.



(Coddington et al., 2016; Ideguchi, 2017). As shown in Figure 1, two combs with nominal repetition
rates of $f_r$ and offset by $\Delta f_r$ are phase locked together, transmitted through a sample, and their
heterodyne signal measured on a photodetector. The resulting rf frequency comb can be mapped back
to the optical domain to generate an overall spectrum, as shown in Figure 1(b), that is the product of the
comb spectra and any atmospheric absorption. One important difference between DCS and other
broad-band laser techniques is that here all wavelengths are measured at once rather than sequentially
as would be the case for a swept laser system; as a result, DCS is much more immune to spectral
distortions from turbulence effects. Moreover, for a fully phase-locked comb, as is used here, the optical
frequency axis is stable and known to high accuracy, and the instrument lineshape is effectively the sum
of two delta-functions, as shown in the spectrum in Figure 1(b). Alternatively, DCS can be thought of as
high-resolution Fourier-transform spectroscopy with diffraction-limited light sources, no moving parts,
negligible instrument line shape, and a rapid scanning rate of $1/\Delta f_r$, which we tune to be faster than
turbulence-induced intensity variations. Here, both combs are transmitted over the open path yielding
the atmospheric absorption spectrum, but it is also possible to transmit only a single comb through the
air to measure both dispersion and absorbance (Giorgetta et al., 2015; Coddington et al., 2016).
Figure 2 provides an overview of our experiment. Two DCS instruments measured the
atmospheric absorption across a 2-km-roundtrip open path that extended from the top of a building at
the National Institute of Standards and Technology (NIST) Boulder campus to a pair of retroreflectors
located on a nearby hill. Both DCS instruments were based on a similar overall design and used self-
referenced, stabilized frequency combs (Sinclair et al., 2015), but one was built by a team at NIST and
the other by a team at the University of Colorado; they are hereafter referred to as DCS A and DCS B,
respectively. As outlined below, the two instruments differed in their exact design and physical
parameters. Nevertheless, no instrument-specific calibration or bias offset was applied to either system.
The acquired atmospheric absorption spectra were fit to retrieve the column density of $CO_2$, $CH_4$, and
$H_2O$ (as well as HDO and $^{13}CO_2$ at lower precision) along with the path-averaged temperature from the
$CO_2$ spectrum. From these data, combined with the measured atmospheric pressure and the path length
(measured via time-of-flight laser ranging), we retrieved the path-averaged dry mole fractions as a
function of time, which are compared between DCS instruments and to a nearby cavity ringdown (CRDS)
point sensor.
2.2 Dual comb spectrometer
Figure 3(a) shows a simplified schematic of both DCS setups. Briefly, each DCS system used two
mutually coherent self-referenced erbium-doped fiber frequency combs based on the design of (Sinclair
et al., 2015) with nominal repetitions rates and $\Delta f_r$ given in Table 1. Mutual optical coherence between
the combs is enforced by phase-locking an optical tooth of each to a common cw laser and the carrier-
envelope offset frequency of each to a common quartz microwave oscillator. Absolute frequency
accuracy is then enforced by a bootstrapped approach that effectively locks the common cw laser to the
same quartz microwave oscillator (Truong et al., 2016). The result is sub-Hz mutual coherence, ~120-kHz
absolute linewidths, and $3.6 \times 10^{-5}$ cm$^{-1}$ absolute frequency accuracy (Truong et al., 2016). This linewidth
is orders of magnitude lower than the ~5 GHz or ~0.2 cm$^{-1}$ width of pressure-broadened absorption
lines. The direct output of the combs is spectrally broadened in highly nonlinear fiber to cover 7140-
5710 cm$^{-1}$ (1.4-1.75 µm) and then filtered to isolate the spectral region of interest from 6021 to 6388
cm$^{-1}$ (1565 to 1661 nm).
The combined light from both combs is transmitted via single-mode fiber to a telescope, where
it is launched to a retroreflector. The returning signal is collected onto an amplified, 100-MHz-
bandwidth InGaAs photodetector and digitized at $f_r$. We acquire a single interferogram at a period of
$1/\Delta f_r$ or 1.6 ms for DCS A; 100 such interferograms are directly summed in real time on a field-
programmable gate array (FPGA). These are transferred to a computer where they are carrier-phase





corrected and further summed over an acquisition time of ~30 seconds. These summed interferograms
are then Fourier transformed and scaled, using the known optical frequency comb tooth positions, to
generate a transmission spectrum (e.g. Figure 4a) spanning 367 $cm^{-1}$ (>10 THz) with a point spacing of
0.0067 $cm^{-1}$.

The exact optical layout of DCS A is given in (Truong et al., 2016). While following the same
basic design, DCS B differs in several technical details. These include a slightly different output
spectrum, as well as slight different comb tooth spacings and offset frequency, minor differences in the
reference cw laser and its locking scheme, and different amplifier design, launched and received powers,
and telescope design. Some of these differences are laid out in Figure 3, Table 1, and Section 2.3 below.

We have found that the use of stabilized, phase coherent frequency combs is a necessary but
not sufficient prerequisite to reaching sub-percent agreement in retrieved gas concentrations. It is
critical that the spectrally-filtered comb output does not include stray unfiltered light. Similarly, any
stray reflections from the telescope that can "short circuit" the atmospheric path must be avoided. As
with FTIR systems, nonlinearities are problematic. In the optical domain, nonlinearities can arise when
the combs are combined in fiber with high optical power. These are minimized for DCS A by filtering the
light, which decreases the peak powers, before combining the combs. For DCS B the combs do not have
a booster amplifier and thus have significantly lower power. Nonlinearities in the photodetection can
also occur (Zolot et al., 2013); in laboratory tests with a CO reference cell, we verified no bias in
retrieved concentration as a function of received power up to 300 µW, which is a factor of two higher
than the maximum power for the open path data. It was also critical to match the interferogram
amplitude to the full dynamic range of the analog-to-digital converters (ADCs) to avoid effective
nonlinearities in the digitization process.
2.3 Launch/Receive telescope
The two telescope systems are shown in Figure 3(a). Due to the large spectral bandwidth,
reflective optics are preferred to minimize chromatic dispersion. For DCS A, the launch/receive system
was based on a bi-directional off-axis parabolic telescope with a 3" aperture while for DCS B, it was
based on a 6"-aperture Ritchey-Chretien (RC) telescope with the light launched separately from behind
the secondary mirror. In both cases, the launched beam diameter was ~ 40 mm and the light was
directed to a hollow corner-cube retroreflector of 2.5" (DCS A) or 5" (DCS B) diameter. A slow servo was
implemented for long-term pointing of the telescope to the retroreflectors. For this servo, a low-
divergence 850 nm LED is co-aligned with the telescope and its retro-reflected light is detected by a co-
aligned CMOS camera with a long focal-length lens and an 850-nm-bandpass optical filter. We then
servo the overall telescope pointing via its gimbal using the LED spot location on the camera. Further
servo details are described in Cossel et al. (2017).
Figure 3(b) shows the return power for both systems as a function of time. For reference, the
minimum return power required to obtain useful spectra was ~15 µW (horizontal black line). At lower
powers, the acquired individual spectra are excluded. Turbulence-induced intensity variations are lower
for the RC-telescope than the off-axis parabolic telescope because of its larger aperture; however, the
long-term stability of the off-axis parabolic telescope was better due to a higher-quality gimbal system.
The collection efficiency of the 6" RC telescope system was about 10-20% in low to moderate turbulence
($C_n^2$ of $10^{-14}$). The collection efficiency of the off-axis parabolic telescope system was lower, at ~ 2-4% in
similar conditions, due to 1) the smaller collection aperture and 2) the 50:50 beam splitter, which causes
a factor of 4 loss. Attempts to replace the 50:50 splitter with a polarizing beam splitter and quarter-
wave plate combination increased the collection efficiency but introduced additional etalons across the
spectrum and for this reason was not used.
2.4 Data processing



The acquired transmission spectra are the product $S(v) = I_0(v) \times e^{-A(v)}$, where $I_0$ is the
geometric mean of the two individual comb spectra, $A(v)$ is the desired atmospheric absorbance, and $v$
is the average optical frequency of the two participating comb teeth, (e.g. Fig. 1(b)). We fit the natural
logarithm of the transmission spectra, $-\ln[S(v)] = -\ln[I_0(v)] + A(v)$, where the first term is
represented by a piecewise polynomial and the second by an absorption spectrum calculated from a
spectral database with floated concentrations of $^{12}CO_2$, $^{13}CO_2$, $^{12}CH_4$, $^{13}CH_4$, $H_2O$, and HDO. For a spectral
database we use HITRAN 2008 (Rothman et al., 2009) and Voigt lineshapes as this generates a consistent
set of line parameters across our conditions and gases. The fit is performed in three steps: first, we fit
the polynomial (typically 7th order) over small windows (typically 100 GHz or 3.33 cm$^{-1}$) and include the
expected absorbance from relevant gas absorption lines. These polynomials are then stitched together
to generate the overall polynomial baseline, which is removed from the measured spectrum to
find $A(v)$. We then fit only the 30013 ← 00001 $CO_2$ band in order to retrieve the path-averaged
temperature. Finally, $A(v)$ is then re-fit over the entire spectral window by floating the gas
concentrations at the retrieved path-averaged temperature. The retrieved path-averaged
concentrations are converted to wet mole fractions by normalizing to the total number density of air
molecules, which is calculated from the fitted (or separately measured) air temperature combined with
the atmospheric pressure, as measured by a sensor co-located with the CRDS sensor and corrected for
the altitude difference. Finally, wet $CO_2$, $^{13}CO_2$, and $CH_4$ are converted to dry values ($XCO_2$, $X^{13}CO_2$, $XCH_4$)
using XS = S/(1-$c_{H2O}$) where XS is the dry species concentration, S is the retrieved wet species
concentration and $c_{H2O}$ is the retrieved $H_2O$ volume mole fraction.
3 Intercomparison Results and Discussion
3.1 Atmospheric spectrum comparison
Figure 4(a) shows the overall raw DCS transmission spectra from the two instruments averaged for a
three-hour period. They differ significantly because of the different comb intensity profiles, $I_0(v)$.
However, after the polynomial baseline fit discussed above is applied, the resulting 3-hour averaged
absorption spectra are nearly identical as shown in Figure 4(b). The inset of Figure 4(b) shows the data
sampling points (spaced at ~200 MHz) across several absorption lines with width of 5 GHz 0.2 cm$^{-1}$)
indicating we have sufficient optical resolution to over-sample the lines. The difference of the
absorption spectra, shown as the black line in Figure 4(c), has a standard deviation of 9×10$^{-4}$ with no
observable structure at absorption lines. This difference is dominated by an etalon on the off-axis
telescope used with DCS A. After manually fitting out the etalon structure, DCS A and DCS B agree to
better than 5×10$^{-4}$ (limited by the instrumental noise level) over the full spectral region (with the
exception of a 7 cm$^{-1}$ section at 6290 cm$^{-1}$), and better than 2.5×10$^{-4}$ over the region near 6100 cm$^{-1}$
where both DCS systems have significant returned optical power. This very high level of agreement
between the two spectra shows that there are no instrumental line shapes or detector nonlinearity
effects distorting the observed spectral line shapes; otherwise, structure would be observed in the
difference. Thus, the two DCS instruments measure the same comb-tooth-resolved atmospheric
absorbance spectrum.
Figure 4(d) shows the residuals after fitting the absorption lines in the DCS A spectrum to HITRAN
2008 and removing the etalon. The higher SNR of the DCS A yields an even lower broadband noise than
the difference spectrum, but there are clear residuals near spectral lines attributable to incorrect line
shapes/parameters in the HITRAN 2008 database. Nevertheless, the overall magnitude of the residuals
is very small in comparison to the spectral absorption.
3.2 Comparison of retrieved mole fractions from DCS A and DCS B
From the fitted concentrations, we retrieve the mole fractions as outlined in Section 2.4. The
retrieved time series for $XCO_2$, $X^{13}CO_2$, $XCH_4$, $H_2O$, and HDO are given in Figure 5 at ~30 second intervals.





Gaps in the data are due to either telescope misalignment (primarily on the 6" RC telescope due to the
lower-quality gimbal system) or, more rarely, a loss of phase lock of one of the four frequency combs.
Excellent agreement is observed between both systems for all retrieved concentrations. Figure 6 shows
the concentration differences, which exhibit a high-frequency white noise consistent with the
quadrature sum of the DCS precisions given in Section 3.3. In addition, the differences show a slow
wander about zero indicating slowly changing, small offsets between the two DCS instruments. $CH_4$ also
shows a small negative offset for the second week of the campaign. A Gaussian curve approximates the
distribution of the differences over the full two weeks reasonably well and is shown in Figure 7. At 32-
second averaging times, the mean and width of the distributions are $\Delta XCO_2 = 0.57 \pm 2.4$ ppm, $\Delta XCH_4$
$= -7.0 \pm 16$ ppb, $\Delta c_{H2O} = 36 \pm 90$ ppm, and $\Delta c_{HDO} = 390 \pm 860$ ppm. These widths decrease to 1.5
ppm, 12 ppb, 66 ppm, and 480 ppm, respectively, for 5-minute averaging times. These mean values
correspond to a relative offset of 0.14 % $CO_2$, -0.35 % $CH_4$, and 0.4 % $H_2O$ and are close to the WMO
compatibility standards of 0.1 ppm for $CO_2$ and 2 ppb for $CH_4$ (Tans and Zellweger, 2015). We emphasize
the agreement here is achieved over a two-week period despite outdoor temperature variations of 4.6
to 28.9 °C, DCS instrument ambient temperature variations from 17 to 25 °C, 10% to 90% relative
humidity fluctuations, and large turbulence-induced return power fluctuations.
Table 2 summarizes the systematic uncertainties of the DCS systems. The choice of spectral
model effectively sets the calibration that converts the measured absorbance spectrum to path-
averaged concentrations. The temperature primarily affects the conversion of the path-averaged
concentration to mole fractions (through the calculation of the overall air concentration). For the direct
intercomparison, both DCS data were analyzed with a common spectral model (HITRAN 2008) and
temperature in order to separate out instrument-specific systematics from the more fundamental
connection between absorption and concentration. Below we discuss these instrument-specific
systematics (given in the top part of Table 2). A discussion of the uncertainties from the spectral model
and temperature (given in the bottom part of Table 2) is given in Section 3.5.
To explore the source of the small systematic offsets between the DCS retrievals, we have
performed a number of control comparisons. In the processing, we have varied the initial concentration
guess in the fit with negligible effect. We have also varied the polynomial baseline fit by adjusting the
window size from 100 to 150 GHz and polynomial order from $7^{th}$ to $9^{th}$ order and again found negligible
variations of 0.02 % for $CO_2$ (<0.07 ppm), 0.07% for $CH_4$ (<1.4 ppb), and 0.05% (~4 ppm) for $H_2O$. In
laboratory tests, we verified that the two DCS instruments retrieve the same $CO_2$ concentrations to
within 0.04% for 8450 ppm of $CO_2$ in a 30-meter multipass cell (roughly mimicking the total absorption
over the open path). In open-path tests, we have separated effects of the detection/acquisition system
and optical system. First, the detected DCS A return signal was split to the two separate data acquisition
systems. The two processed signals yielded small differences of 0.16 ppm $CO_2$, 0.34 ppb $CH_4$, and 1.0
ppm $H_2O$, presumably due to residual nonlinearities and reflections in the rf system and digitization.
Second, the outgoing DCS A comb light was split and directed to the two different telescopes and
acquisition systems. These two processed signals yielded larger differences of 0.45 ppm $CO_2$, 1.5 ppb
$CH_4$, and 56 ppm $H_2O$, possibly due to scattered light or polarization dependences in the launch and
receive optical systems. Finally, residual phase noise between the two combs in a single DCS system can
cause small biases in the retrieved concentrations, but these should be well below 0.1% in this
configuration (Truong et al. 2017, in prep). All these instrument uncertainties are summarized in Table 2.
3.3 DCS precision
Figure 8 shows the precision versus averaging time (determined using the modified Allan
deviation) based on the scatter across a 6-hour period over which the $CO_2$ and $CH_4$ concentrations are
reasonably flat, shown as the highlighted part of Figure 5. (The Allan deviation for $H_2O$ is not calculated
because the atmospheric $H_2O$ concentration varies significantly over this time period.) Under perfectly



stable concentrations and white instrument noise, the precision should decrease as the square root of
averaging time, indicated as a grey line in Figure 8. Initially, the Allan deviations do follow this slope, but
the atmospheric concentrations, especially of $CO_2$, vary over this 6-hour period and the Allan deviations
reach a floor at ~ 1000 s.
The precision at 30-second and 5-minute averaging time is given at the bottom of Table 1.  DCS
A has superior $CO_2$ precision because it has higher received optical comb power in that spectral region,
whereas the DCS instruments have similar received power in the $CH_4$ spectral region and therefore
similar $CH_4$ precisions. Regardless, the precision of either instrument is sufficiently high to measure the
characteristic atmospheric fluctuations of these gases on tens-of-seconds timescales.
3.4 Comparison of open-path DCS to a cavity ringdown point sensor (CRDS)
A commercial cavity-ringdown point sensor, Picarro Model 1301[2] (Crosson, 2008), was also located
along the path as shown in Figure 2. Its inlet was 30 m above ground on a radio tower, approximately
160 m perpendicular to the DCS beam path. Figure 9 compares the DCS A and CRDS (smoothed to 32-s
resolution) time series. In general, their overall shapes agree well with both systems tracking ~40 ppm
variations in $XCO_2$, 200 ppb variations in $XCH_4$, and 1% variations in $H_2O$ over days. Nevertheless, there
are clear discrepancies in terms of both short-duration spikes and a long-term overall offset between
the DCS and CRDS time series.
The short-duration spikes are present in the CRDS time series and presumably arise from the very
different spatial sampling of the two instruments. The DCS system measures the integrated column over
one kilometer (one way), while the CRDS is a point sensor and therefore much more sensitive to local
sources. For example, a 1 $m^3$ volume of air containing 500 ppm of $CO_2$ from a vehicle driving under the
sampling line will result in a sharp spike in the CRDS data as the air mass passes the sampling inlet.
However, that same air mass will result in only a 0.025% or 0.1 ppm increase in the DCS path-averaged
concentration (assuming a 400 ppm background). These spikes in the CRDS time series are damped here
by the 32-second smoothing but are occasionally evident especially during the second week. The general
scarcity of such events does suggest that the air over the open path is usually fairly well mixed.
The long-term overall offset between the CRDS and DCS data is a consequence of their very different
calibrations. The CRDS is tied to the WMO scale for $CO_2$ and $CH_4$ by directly injecting known dry WMO-
calibrated $CO_2$/$CH_4$ mixtures at different trace gas concentrations and different water vapor
concentrations into its temperature- and pressure-controlled sampling cavity.  This instrument was
calibrated shortly after the measurement campaign and should thus have an absolute uncertainty close
to that of the WMO-scale uncertainties of ~ 0.07 ppm for $CO_2$ (Zhao and Tans, 2006) and ~1.5 ppb for
$CH_4$ (Dlugokencky et al., 2005).  In contrast, the DCS has no instrument-specific calibration but relies
completely on a fit to a spectral database to extract the gas concentrations from the measured
absorbance across a wide range of ambient pressures and temperatures. Here, we use HITRAN 2008
which has $^{12}CO_2$ linestrength uncertainties of 1-2 %, $^{12}CH_4$ linestrength uncertainties of 10-20%, and
$H_2^{16}O$ linestrength uncertainties of 5-10% (Rothman et al., 2009), leading to a poorer absolute
calibration than the WMO-calibrated point sensor. From the data in Figure 9, the differences between
the CRDS and DCS data across the two-week period are -3.4 ± 3.4 ppm $CO_2$, 17 ± 15 ppb $CH_4$, and 580 ±
462 ppm $H_2O$ at 5-minute averaging. These correspond to relative offsets of -0.85% for $CO_2$, 0.94% for
$CH_4$, and 6.9% for $H_2O$, well within the stated uncertainties of HITRAN 2008.  In previous DCS
measurements, we found slightly different offsets, specifically 1.78% for $CO_2$, 0.20% for $CH_4$, and 1.74%
for $H_2O$ in (Rieker et al., 2014) and ~1 % for $CO_2$ in (Giorgetta et al., 2015).  However, these previous data

---

[2] The use of trade names is necessary to specify the experimental results and does not imply endorsement by the
National Institute of Standards and Technology.



covered much shorter timespans, used an older CRDS point sensor calibration, and may have included
small systematic offsets in the DCS systems due to technical issues discussed in Section 3.2.
This basic discrepancy between retrievals based on lineshape parameters from a spectral database
and manometric calibrations (WMO standard) is not unique to DCS.  Several studies have calibrated the
Total Carbon Column Observing Network (TCCON) retrievals against WMO-based instruments (Wunch et
al., 2010; Messerschmidt et al., 2011; Geibel et al., 2012; Tanaka et al., 2012). Although TCCON is not a
solely HITRAN-based analysis (Wunch et al., 2011), a correction factor of 0.9898 for $CO_2$, 0.9765 for $CH_4$,
and 1.0183 for $H_2O$ (Wunch et al., 2010) is needed to bring the overall TCCON retrievals into agreement
with the WMO-based data. Additionally, theoretical calculations by (Zak et al., 2016) found an
approximately 0.5% difference between $CO_2$ line parameters from HITRAN 2012 and their density
functional theory calculations and an additional 0.5% difference between the calculations and new
measurements by (Devi et al., 2016) in the 1.6-micron region.  Certainly this discrepancy between
retrievals from HITRAN and WMO-calibrated instruments is not fundamental and further experimental
work should lead to improved spectral database parameters and much better agreement. As noted in
earlier work on $CO_2$, it will be important to establish both the correct linestrengths as well as account for
complex lineshapes and line mixing (e.g. Devi et al., 2007; Thompson et al., 2012; Bui et al., 2014; Long
et al., 2015; Devi et al., 2016). A direct comparison of the open-path DCS spectra acquired here and
laboratory DCS spectra acquired for WMO-calibrated gas samples can contribute to these future
improvements and is planned. Finally, we emphasize that because the DCS instruments record the
atmospheric absorption without instrument distortions, as spectral models improve, past open-path
spectra can be refit with reduced uncertainty.
An accurate path-averaged air temperature is also important to avoid systematic offsets.  Unlike
vertical total-column measurements through the entire atmosphere, km-scale open horizontal paths
should have relatively low temperature inhomogeneities of around a few degrees C, and thus the use of
a single "path averaged" temperature in the fit is sufficient for accurate retrievals. We verified this
through a sensitivity study comparing retrievals for simulated spectra with temperature gradients up to
10 °C over the path; the resulting bias was below 0.03 ppm $CO_2$ (0.007%) and 0.4 ppb $CH_4$ (0.022%), as
shown in Table 2.  On the other hand, any error in the path-averaged temperature can bias the mole
fractions through two effects. First, the retrieved path-averaged concentration will vary weakly with
temperature because of temperature-dependent line parameters. Second and dominantly, the final
mole fraction calculation requires normalization by the air density. Here, this density is calculated from
the ideal gas law using the measured air pressure and path-averaged temperature. Therefore, a
fractional error in temperature leads to a corresponding fractional error in mole fraction. For example, a
0.15% uncertainty in mole fraction requires 0.5 °C uncertainty in the path-averaged air temperature.
(See Table 2.) We verified that this simple linear relationship is valid up to a temperature uncertainty of
10 °C in a sensitivity study. From the discussion in Appendix A, the use of a point temperature sensor
near the end of the open path is clearly insufficient to achieve <0.5 °C uncertainty at many times of the
day. Instead, for the data here, we have used the fitted path-averaged temperature, as discussed in
Section 2.4. The approach effectively relies on the spectral database but, in this case, on the variation in
the Boltzmann distribution of the J-level population with temperature. In Table 2, we have taken a
hopefully conservative uncertainty of 0.5 °C for the path-averaged temperature, but more work is
needed to establish the true uncertainty from these retrieved values. Finally, we note the fractional
uncertainty in the measured atmospheric pressure from the sensor or altitude-based pressure changes
across the optical path was below 0.36%.
4.  Diurnal cycles and source analysis
The two weeks of open path data are analyzed for diurnal cycles, as shown in Figure 10 with the intent
of an initial understanding of $CO_2$ and $CH_4$ sources.  For this analysis, the wind speed and wind direction





were taken from the NCAR Mesa weather data (ftp://ftp.eol.ucar.edu/pub/archive/weather/mesa/),
while the gas concentrations are from DCS A.
4.1 Carbon dioxide
As expected, the median of the diurnal cycle for $CO_2$ shows a peak in the early to mid-morning
from commuter traffic after which the $CO_2$ concentration decreases as the boundary layer rises.  It
remains approximately steady throughout the afternoon, decreases to a minimum between 19:00 and
20:00, and then increases slightly overnight as the boundary layer collapses.  We hypothesize that the
afternoon behavior is due to the change in wind direction.  Often overnight and through early morning
the wind blows from the west to southwest, which brings in cleaner background air from the mountains
bordering Boulder. However, in late morning the predominant wind direction shifts to the east and
southeast, possibly bringing in higher $CO_2$ concentrations from the Denver metropolitan area – which
lies approximately 30 km to the southeast of Boulder – over the course of the afternoon. Typically, the
evening wind shifts back to out of the west, once again bringing in the cleaner mountain air and with it a
decrease in $CO_2$ concentration.
4.2 Methane
Methane has a significantly weaker diurnal cycle than carbon dioxide, which is consistent with a
species that lacks significant diurnally-varying local sources. Rather, its concentration follows expected
variations in the boundary layer height; the concentration increases overnight into the early morning as
the boundary layer collapses, and then decreases during the late morning through afternoon as the
boundary layer rises again. The largest likely methane source near Boulder is local oil/gas fields, but
these typically lie to the northeast, while the wind directions are generally out of the west to southeast.
It is also possible that the methane comes from leaking natural gas infrastructure within the city.
5 Conclusions
Here we provide the first quantitative comparison of open-path dual comb spectroscopy
instruments.  The dual-comb spectrometers were based on fully phase-coherent and stabilized fiber
frequency combs and operated nearly continuously over a two-week period. We performed these
measurements over adjacent 2-kilometer round-trip paths to measure concentrations of dry $CO_2$, dry
$CH_4$, $H_2O$, and HDO.  The measured atmospheric absorbance spectra agree to better than $10^{-3}$.
Correspondingly, we find excellent agreement between the retrieved concentrations from the two
instruments without the need for instrument calibration:  over two weeks of near-continuous
measurements, the retrieved $CO_2$ concentrations agree to better than 0.14% (0.57 ppm), $CH_4$
concentrations agrees to better than 0.35% (7.0 ppb), and $H_2O$ concentrations agrees to better than
0.4% (36 ppm).  These values are very close to the WMO compatibility goals.  The remaining
disagreement is likely due to scattered stray light, polarization dependencies, and residual comb phase
noise.  We further compare the DCS measurements to a cavity ringdown point sensor located along our
path.  The measured dry $CO_2$ mole fraction agrees to within 1%, the $CH_4$ dry mole fraction to within
1.2%, and $H_2O$ mole fraction to within 6.2%.  However, this CRDS point sensor is directly calibrated to
the WMO scale for $CO_2$ and $CH_4$ while the DCS results are based on HITRAN 2008; we attribute the
disagreement in $CO_2$ and $CH_4$ to inaccurate line parameters in the HITRAN database. (Most of the water
discrepancy is attributed to the imperfect absolute water calibration of the CRDS point sensor.) Further
improvements to the spectral database should reduce these discrepancies. Finally, this open-path DCS
can exploit even broader spectrum combs up to 2.3 μm and down to 1.1 μm (Zolot et al., 2012; Okubo et
al., 2015), which would enable measurements of similar quality for $^{13}CO_2$, $NH_3$, $N_2O$, and $O_2$. These
results make open-path DCS a promising new system for greenhouse gas flux measurements from
distributed sources.



The authors declare that they have no conflict of interest.
Acknowledgements:  We thank T. Newburger and K. McKain for assistance with the CRDS calibrations, T.
Bullet and J. Kofler for assistance in setting up the CRDS sampling, and R. Thalman for detailed long-path
instrument correlation data, and A.J. Fleischer and A. Karion for assistance with the manuscript.  This
work was funded by the Defense Advanced Research Program Agency DSO SCOUT program, ARPA-E
MONITOR program under Award Number DE-AR0000539, and the NIST Greenhouse Gas and Climate
Science Initiative.  EMW and KCC are supported by National Research Council postdoctoral fellowships.

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



| | DCS A | DCS B |
|---|---|---|
| **Design Details** | | |
| Comb 1 repetition rate ($f_r$) | ~200 MHz | ~204 MHz |
| Difference in repetition rate ($\Delta f_r$) | 624 Hz | 870 Hz |
| Spectral filtering | Before combining combs | After combining combs |
| Booster amplifier | Yes | No |
| Average power launched | 4 mW | 1.5 mW |
| Filtered spectral output | 6376 to 6023 cm$^{-1}$ | 6359 to 6003 cm$^{-1}$ |
| Telescope design | Home-built 3"-diameter off-axis telescope | Modified commercial 6"-diameter Ritchey-Chretien telescope |
| Retroreflector | 2.5" HCC, 5 arc seconds | 5" HCC, 5 arc seconds |
| Round-trip path length | 1950.17 m | 1963.67 m |
| Typical averaging time | 32 s | 28 s |
| **Performance Metrics** | | |
| 30-second precision | 0.90 ppm $CO_2$, 9.6 ppb $CH_4$ | 2.15 ppm $CO_2$, 11.5 ppb $CH_4$ |
| 5-minute precision | 0.24 ppm $CO_2$, 2.1 ppb $CH_4$ | 0.60 ppm $CO_2$, 3.2 ppb $CH_4$ |

**Table 1:** Specifications of the two DCS systems. HCC: hollow corner cube





| Systematic source [effect] | Effect on retrieved $CO_2$ | Effect on retrieved $CH_4$ | Effect on retrieved $H_2O$ |
|---|---|---|---|
| Fitting procedure [initial guess, baseline polynomial order and window size] | 0.07 ppm | 1.4 ppb | 4 ppm |
| Rf detection and processing [rf reflections, ADC nonlinearities] | 0.16 ppm | 0.34 ppb | 1.0 ppm |
| Telescope system [Scattered light, polarization effects] | 0.45 ppm | 1.5 ppb | 56 ppm |
| Spectral database [linestrengths in HITRAN 2008] | 1-2% | 10-20% | 5-10% |
| Temperature path inhomogeneities [if <10 C across path] | 0.024 ppm | 0.36 ppb | 3.52 ppm |
| Path-averaged temperature [for 0.5 C uncertainty] | 0.64 ppm | 2.9 ppb | 8.6 ppm |

**Table 2:** List of systematic uncertainties. See discussion in Section 3.2 and 3.4 for more details. Upper
half of table: systematics due to hardware and software. Lower half of the table: systematics due to
spectral model and temperature uncertainties.




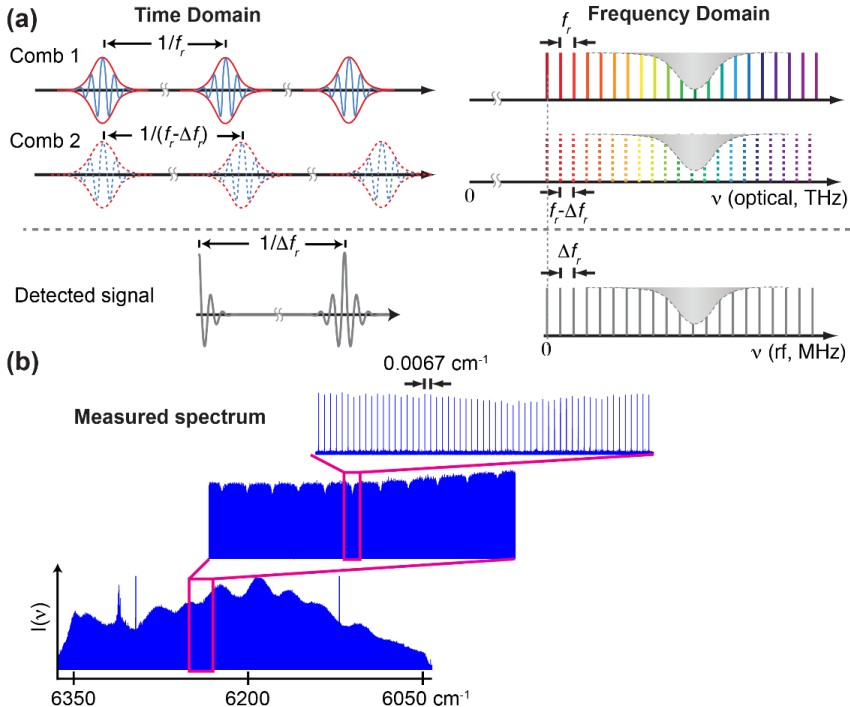

*Figure 1. (a) Time and frequency domain overview of dual comb spectroscopy. Two frequency combs are phase-locked together*
*with pulse repetition rates of $f_r$ and $f_r - \Delta f_r$. Their detected heterodyne signal is a series of interferograms in the time domain, or*
*equivalently a comb in the radio frequency (rf) domain. Provided the combs are sufficiently coherent and Nyquist sampling*
*conditions are met, each rf comb tooth maps to a particular, known pair of optical frequency comb teeth (Coddington et al.,*
*2008). As a result, the optical spectrum can be obtained from the magnitude of each rf comb tooth versus the average optical*
*frequency of the relevant comb tooth pair. (b) Actual DCS spectrum acquired over 1.15 seconds for DCS A after transmission over*
*2-km air path. The overall shape is governed by the comb spectrum but there are narrow absorption dips present from*
*atmospheric gases, as shown in the first expanded view. The second expanded view shows the fully resolved rf comb teeth with*
*time-bandwidth limited widths. The highly resolved nature of these spectral elements translates to a negligible instrument*
*lineshape, set by the narrow comb linewidths. The sample points are separated by 0.0067 cm$^{-1}$ (or $f_r$ =200 MHz). For long-term*
*averaging, we implement coherent co-adding of interferograms that effectively measures the power at the individual rf comb*
*teeth (Coddington et al., 2008)*






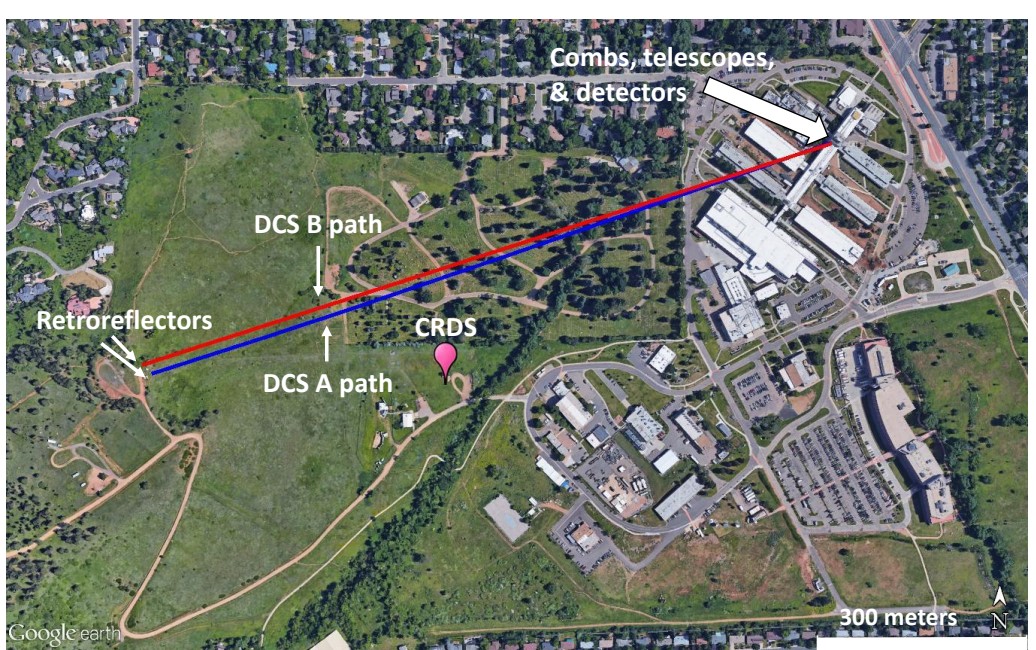

*Figure 2. Setup for the open-path dual-comb spectrometer (DCS) comparison at the NIST Boulder CO campus. The main*
*components for DCS A and DCS B are housed in a rooftop laboratory, including the frequency combs, telescope, receiver, and*
*processor. For each DCS, the combined comb light is launched from a telescope, travels ~1 km through the atmosphere to a*
*retroreflector, and returns to the telescope where it is collected, detected and processed. A separate cavity ringdown point*
*sensor (CRDS) is located nearby with an inlet on a 30-m tower that is located ~160 meters from the nearest point of the free-*
*space DCS paths.*





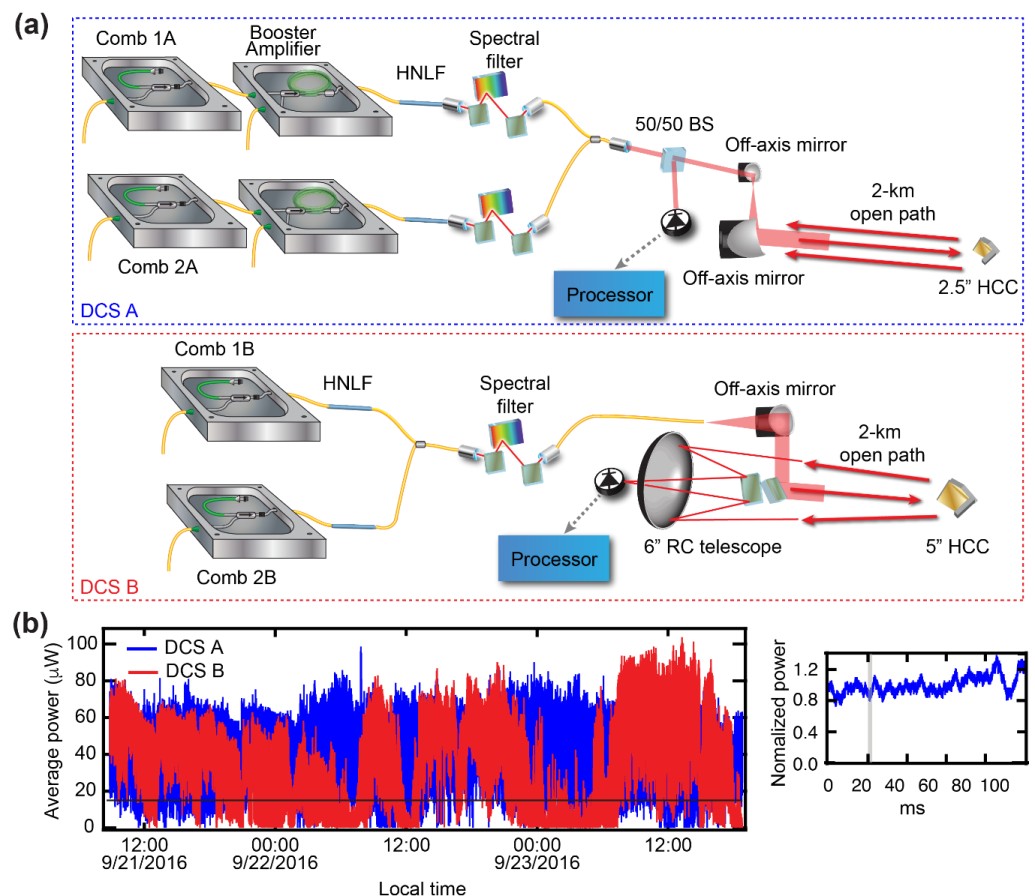

Figure 3. (a) Configuration of DCS A and DCS B, both of which are based on fully self-referenced fiber-laser frequency combs. See
text and Table I for details. DCS A includes a booster amplifier for higher launched optical power than DCS B. (b) Average optical
return power for DCS A (blue) and DCS B (red) measured at the detector over about 2.5 days. The horizontal black line shows the
approximate minimum power for useable SNR (15 µW). Inset: The normalized power fluctuations for DCS A over 100 ms. The
acquisition time for a single DCS spectrum is shown by the thickness of the vertical grey bar. RC: Ritchey-Chretien; HNLF: highly
nonlinear fiber; HCC: hollow corner cube retroreflector; BS: beam splitter.




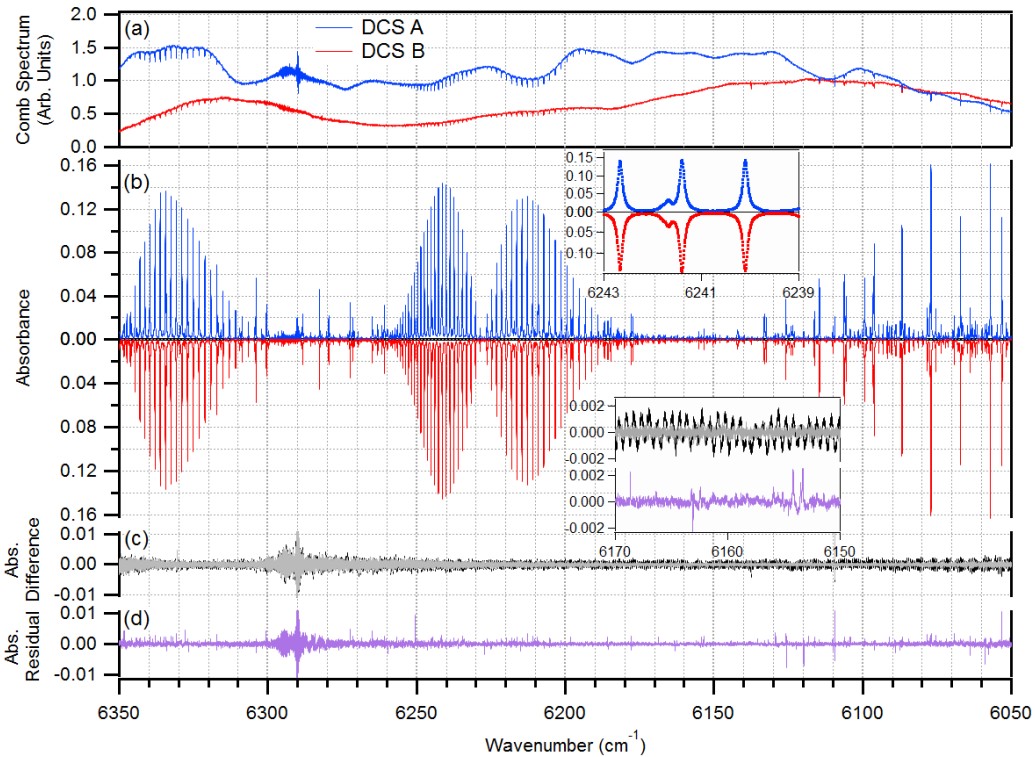


*Figure 4. Raw spectra from DCS A (blue) and DCS B (red). (b) Corresponding baseline-corrected absorption spectra averaged for*
*a three-hour period. The spectra overlap completely on this scale so the DCS A absorbance has been flipped about zero. Inset:*
*expanded view of several $CO_2$ lines. (c) Difference between the absorption spectra from DCS A and DCS B. The difference is*
*shown both before (black trace) and after (grey trace) removing an etalon structure and agree to better than $5 \times 10^{-4}$ after the*
*etalon is removed. Inset: Expanded view. (d) Residuals from a fit of the DCS A spectrum to HITRAN 2008. In general, the residuals*
*are lower noise than the difference spectrum because of the higher signal-to-noise ratio of the DCS A than DCS B, but there are*
*clear structures present near absorption lines due to imperfect line shapes of the spectral database.*





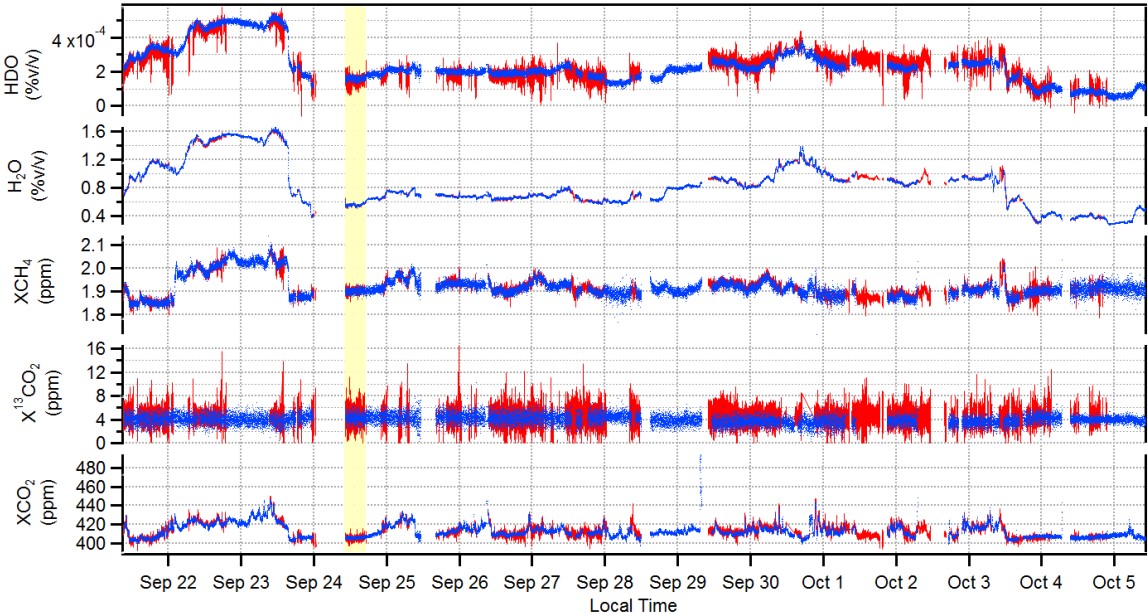

*Figure 5. Concentration retrievals from DCS A (blue dots) and DCS B (red lines) for HDO (% by volume) $H_2O$ (% by volume), dry*
*$CH_4$, dry $CO_2$ and dry $^{13}CO_2$ over two weeks at 30-second intervals. Excellent agreement is observed between both systems for all*
*species, though it is clear that over this path length $^{13}CO_2$ does not provide a strong enough signal to retrieve reliably.*
*Highlighted section: 6-hour, well-mixed period over which Allan deviations (Figure 8) are calculated. Missing data is primarily*
*due to telescope misalignment and less frequently, phase lock by one of the combs.*





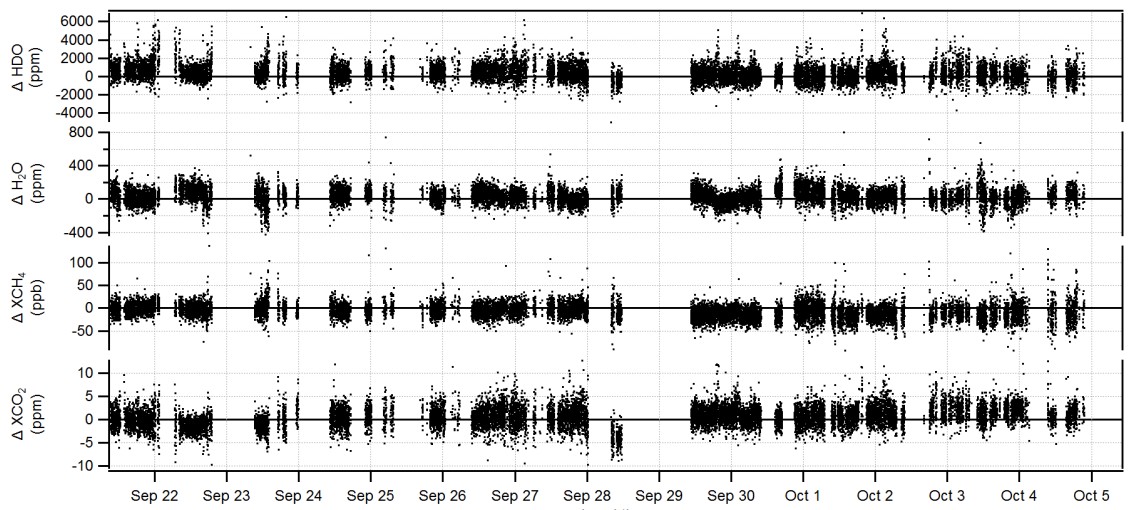

*Figure 6. Time series of concentration differences, where difference is defined as DCS A - DCS B.*





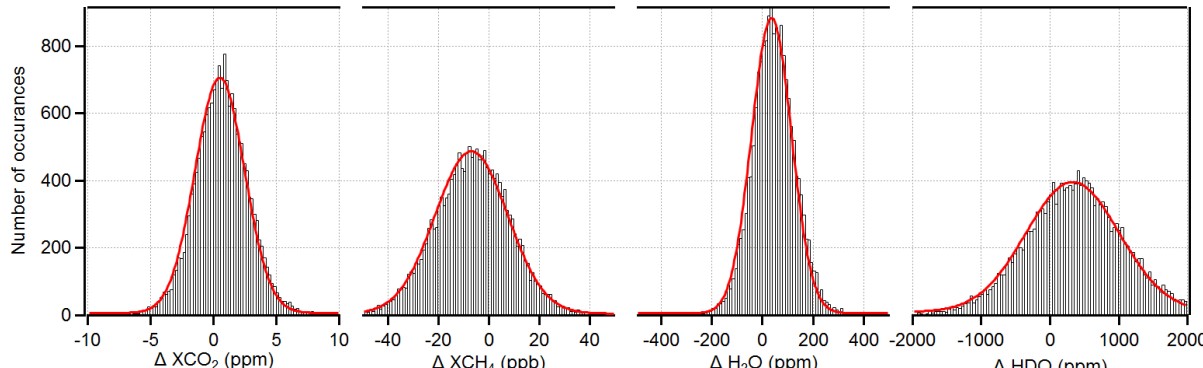

*Figure 7. Statistical distributions of the differences between DCS A and DCS B for dry $CO_2$, dry $CH_4$, $H_2O$, and HDO from Fig. 6. Histograms are shown in black with a fit to Gaussian curves in red. These data are for ~30-second intervals; the widths are approximately halved if the data is averaged to 5-minute intervals.*





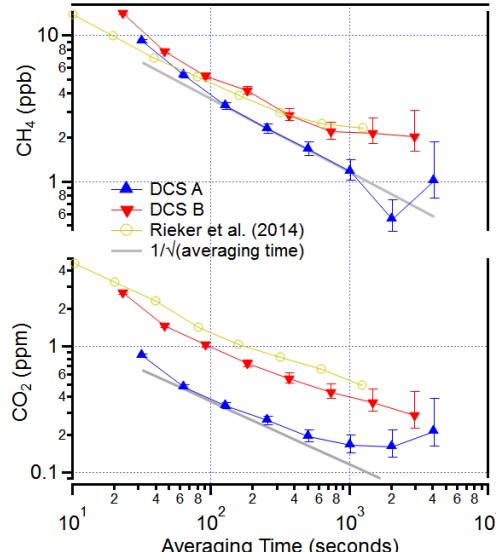

*Figure 8. Precision (Allan deviation) versus averaging time, τ, for CH₄ and CO₂ calculated for DCS A (blue) and DCS B (red) over a*
*2 km path for the time period highlighted in Figure 3. The previously-published precisions from Rieker et al. (2014) are also*
*shown (gold). The grey line illustrates the slope expected for white noise. For DCS A, at averaging times from 30s to 1000s, the*
*precision roughly follows ~40 ppbv/√τ for CH4 and ~4 ppmv/√τ for CO₂ (gray lines).*





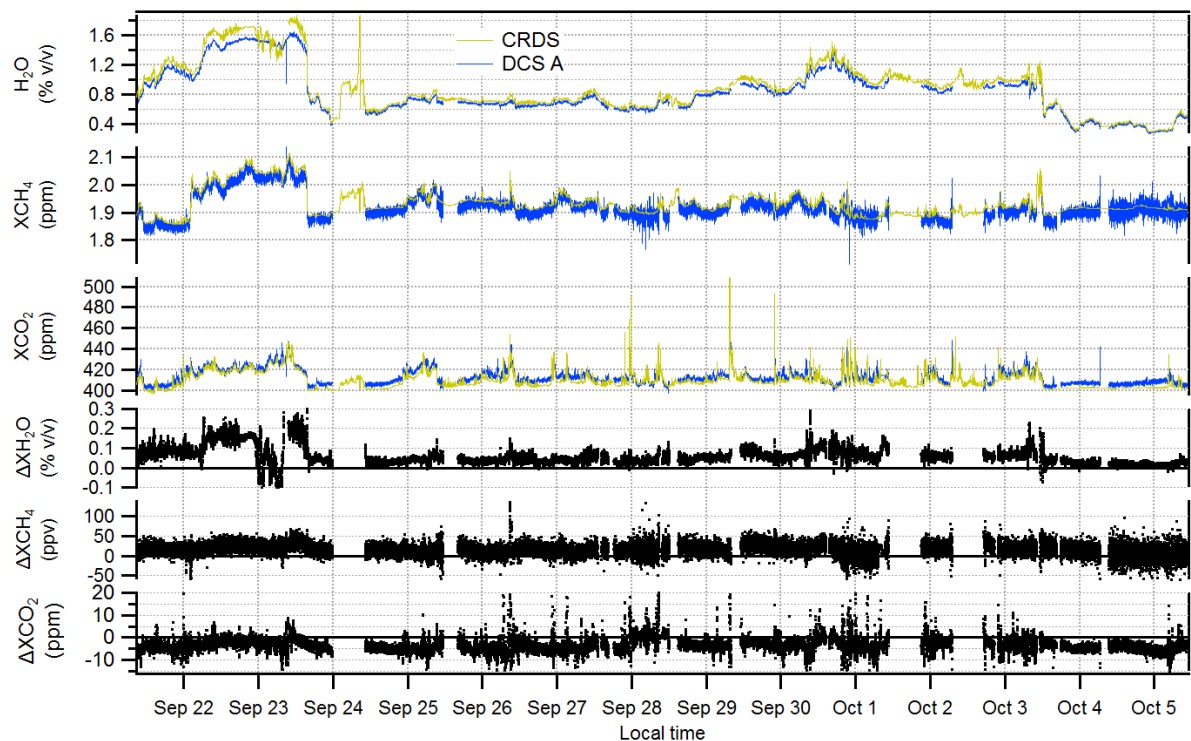

Figure 9. Comparison between the open-path DCS A data (blue) and the point CRDS data (gold) for $H_2O$, dry $CH_4$, and dry $CO_2$ at 32-second intervals over two weeks. The lower three panels directly plot the corresponding difference between the two.





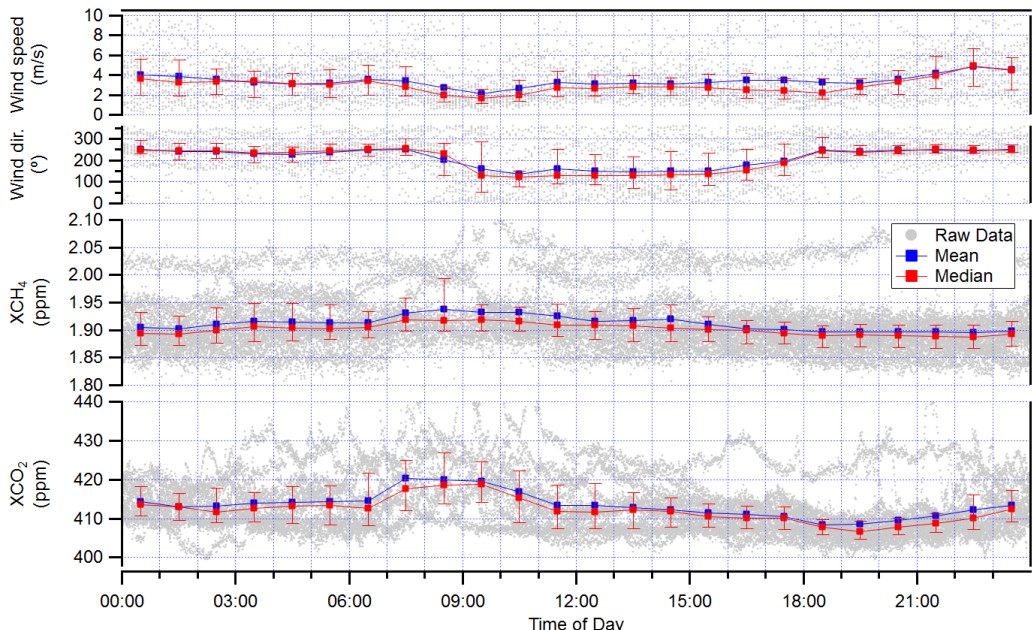

*Figure 10. Diurnal cycles for wind speed, wind direction, XCH₄, and XCO₂. Data from each day in Figure 5 is over-plotted in grey*
*along with the hourly mean (blue) and median (red) values. Uncertainty bars on the median values span the 75th quantile and*
*25th quantile.*



Appendix A:  Temperature studies

As described in Section 2.4, we extract the path-averaged temperature directly from a fit to the 30013
← 00001 overtone band of $CO_2$.  We perform this fit on 5-minute averages, rather than 32-second
averages, under the assumption the temperature changes are still slow at that timescale. This path-
averaged temperature is then used in a subsequent fit over the full spectral region to extract the column
densities, and finally the mole fractions.  We use a common temperature (from DCS A fit) to analyze
both data sets in order to separate out instrument effects from the temperature, but the fitted
temperatures between instruments show less than 0.25 °C bias.
Figure A1 compares this fitted path-averaged temperature from DCS A to three point sensors,
two of which are located on the rooftop near the telescope launch point and one that is located ~ 2.2
km away at an altitude ~200 m above the overall open path.  As shown in Figure A1, the two rooftop
temperature sensors located near the telescope agree well with each other, but do not agree with the
fitted path-averaged temperature. Moreover, that disagreement has a distinct diurnal character,
supporting the argument it arises from a real temperature gradient.  In contrast, the path-averaged
temperature does often agree well with the temperature measured by the third temperature point
sensor located at similar or higher altitude as the open path on the NCAR Mesa building
(ftp://ftp.eol.ucar.edu/pub/archive/weather/mesa/).  These data indicate that the point sensor located
at the telescope site is not a good proxy for the path-averaged temperature; instead, the fitted path-
averaged temperature should be used for the concentration fits because of temperature gradients. Note
that the temperature gradients themselves do not lead to appreciable errors in the retrieved mole
fractions if the correct path-averaged temperature is used (see Table 2 and Section 3.4).


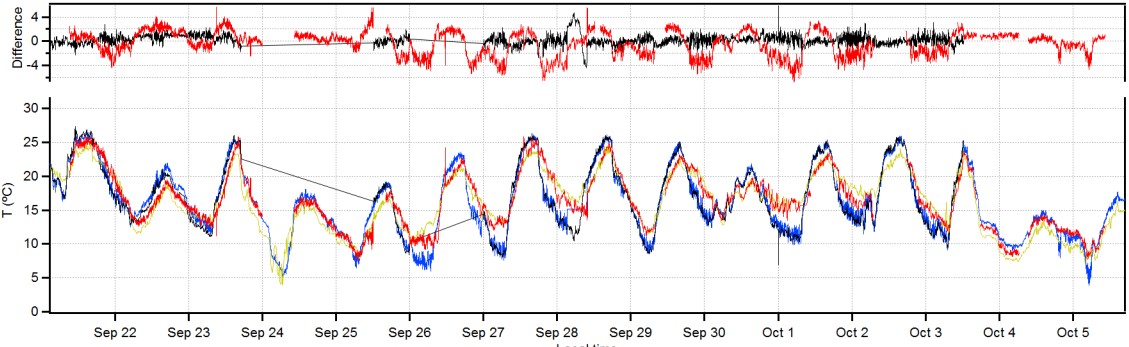


Figure A1:  The fitted path-averaged temperature over two-weeks at 5 minute intervals (red) compared
to the measured air temperature from a roof-top anemometer located near the telescope (blue), a
second thermistor temperature sensor also located on the roof but 100-m distant (black), and a third
rooftop temperature ~2.2 km distant at the NCAR Mesa facility (gold).  Top panel: The difference
between the two rooftop temperature (black) agree to within 1°C, but the difference between these
rooftop sensors and fitted path-averaged temperature (red) shows larger 2-4°C diurnal differences,
indicating it is not sufficient to measure the temperature at one "end point" of the open path. In fact,
the path-averaged temperature agrees better with the more distant, but higher elevation temperature
sensor located at the NCAR Mesa facility.