# Peer review of "Intercomparison of Open-Path Trace Gas Measurements with Two Dual Frequency Comb Spectrometers"

_Atmospheric Measurement Techniques, 2017_

## Referee Comment (RC1) · Anonymous Referee #1 · 24 Apr 2017

This manuscript, entitled "Intercomparison of Open-Path Trace Gas Measurements with Two Dual Frequency Comb Spectrometers," reports on a quantitative evaluation of atmospheric trace gas measurements based on dual-comb spectroscopy. Thanks to their well-polished dual-comb spectrometers and analytical approach, the retrieved dry mole fractions agree to 0.57 ppm for $CO_2$ and 7 ppb for $CH_4$ between the two measurement systems. These results are excellent, while there are some obscure points in the manuscript. Therefore, I recommend the manuscript for publication if following comments and questions are addressed.

[Specific comments]

[1] While I am briefly familiar with the technique of gas spectroscopy, I am not an expert

in atmospheric measurement and concerned about some technical descriptions.

i) Is it OK for AMT readers to use some technical terms such as "WMO-calibration" and "WMO compatibility goal" without any simple explanation?

ii) L207: I was confused with the expressions of concentration. Is it correct that the dry and wet concentrations of carbon dioxide are expressed as "XCO2" and "CO2," respectively?

iii) L209: "Volume mole fraction" might be simply "mole fraction."

iv) Figure 5: I found volume percentage is normally expressed as "v/v% ." Is "%v/v" OK too?

[2] According to the footnote in P2, "ppm" and "ppb" are used for dry concentration (dry mole fraction) and "%v/v" is for wet concentration (wet mole fraction) as in Figure 5. However, "ppm" is used for $\Delta$HDO and $\Delta$H2O in Figure 6 and 7.

[3] I think the explanation about dual-comb spectroscopy is a little insufficient. For example,

i) L94 and 97: Authors should refer to Figure 1(a) here instead of Figure 1 and 1(b).

ii) L102: I could not understand the explanation "the instrument lineshape is effectively the sum of two delta-functions." What does it mean?

iii) L139: Readers might not be sure whether fr is a sampling rate or a bandwidth.

[4] L183: Please define C_n^2.

[5] In Figure 4, the observed HDO is $10^{-4}$ %v/v level, whereas $\Delta$HDO is 1000 ppm level in Figure 6 and 7. Are they consistent?

[6] In the caption of Figure 8, "40 ppmv/$\sqrt{\tau}$" and "4 ppbv/$\sqrt{\tau}$" might be "40 ppm/$\sqrt{(\tau/s)}$" and "4 ppb/$\sqrt{(\tau/s)}$," respectively.

[Technical corrections]

[1] There are some notations without space between the value and unit; for example "1-10s" in L46, "10%" in L51. Please check and correct them.

[2] Notation variability, "dual comb" and "dual-comb."

[3] L94: Reference (Ideguchi, 2017) is missing in the list of references.

[4] L214-215: Notation variability, "three-hour" and "3-hour."

[5] L217: A beginning parenthesis is missing.

Please also note the supplement to this comment:
http://www.atmos-meas-tech-discuss.net/amt-2017-62/amt-2017-62-RC1-supplement.pdf

[Figure]

**Supplement:**

This manuscript, entitled "Intercomparison of Open-Path Trace Gas Measurements with Two Dual Frequency Comb Spectrometers," reports on a quantitative evaluation of atmospheric trace gas measurements based on dual-comb spectroscopy. Thanks to their well-polished dual-comb spectrometers and analytical approach, the retrieved dry mole fractions agree to 0.57 ppm for $CO_2$ and 7 ppb for $CH_4$ between the two measurement systems. These results are excellent, while there are some obscure points in the manuscript. Therefore, I recommend the manuscript for publication if following comments and questions are addressed.

[Specific comments]

1.  While I am briefly familiar with the technique of gas spectroscopy, I am not an expert in atmospheric measurement and concerned about some technical descriptions.

    1)  Is it OK for AMT readers to use some technical terms such as "WMO-calibration" and "WMO compatibility goal" without any simple explanation?

    2)  L207: I was confused with the expressions of concentration. Is it correct that the dry and wet concentrations of carbon dioxide are expressed as "$XCO_2$" and "$CO_2$," respectively?

    3)  L209: "Volume mole fraction" might be simply "mole fraction."

    4)  Figure 5: I found volume percentage is normally expressed as "v/v%." Is "%v/v" OK too?

2.  According to the footnote in P2, "ppm" and "ppb" are used for dry concentration (dry mole fraction) and "%v/v" is for wet concentration (wet mole fraction) as in Figure 5. However, "ppm" is used for ΔHDO and Δ$H_2O$ in Figure 6 and 7.

3.  I think the explanation about dual-comb spectroscopy is a little insufficient. For example,

    1)  L94 and 97: Authors should refer to Figure 1(a) here instead of Figure 1 and 1(b).

    2)  L102: I could not understand the explanation "the instrument lineshape is effectively the sum of two delta-functions." What does it mean?

    3)  L139: Readers might not be sure whether $f_r$ is a sampling rate or a bandwidth.

4.  L183: Please define $C_n^2$.

5.  In Figure 4, the observed HDO is 10-4 %v/v level, whereas ΔHDO is 1000 ppm level in Figure 6 and 7. Are they consistent?

6.  In the caption of Figure 8, "40 ppmv/$\sqrt{\tau}$" and "4 ppbv/$\sqrt{\tau}$" might be "40 ppm/$\sqrt{\tau/s}$" and "4 ppb/$\sqrt{\tau/s}$," respectively.

[Technical corrections]

1.  There are some notations without space between the value and unit; for example "1-10s" in L46, "10%" in L51. Please check and correct them.

2. Notation variability, "dual comb" and "dual-comb."

3. L94: Reference (Ideguchi, 2017) is missing in the list of references.

4. L214-215: Notation variability, "three-hour" and "3-hour."

5. L217: A beginning parenthesis is missing.

---

## Referee Comment (RC2) · Anonymous Referee #2 · 15 Jun 2017

This paper compares the performance of two open-path dual comb spectroscopy (DCS) instruments over two weeks and analyzes the measurement differences, also between the DCSs and a Cavity ring down spectrometer measuring in-situ. The paper describes the function principle of the dual comb spectrometers, the optical setup, and the data processing. The precision of the individual DCS is determined using Allan analysis.

This kind of inter-comparison has not been done so far, and the DCS instruments deployed here are novel. The descriptions and analyses are clear structured. The reviewer recommends the publication of this paper, if the following items are addressed.

[Figure]

General comments:

1. As the authors noted, the dual comb spectrometer has negligible line width (120 kHz) and high spectral resolution, the absorbance resolution is below $10^{-3}$ ($5 \cdot 10^{-4}$), which is limited by the instrument noise. However, the laser spectroscopic instrument typically has an absorbance resolution of $10^{-5}$, limited by the shot noise of the photodiode (also depends on the incoming light intensity). Could the authors be more specific about the origin of the noise? What kind of instrumental noises are these, photodetector noise, noise from the dual comb laser?

2. What benefits in the authors' opinion could be gained from the open-path DCS techniques compared to open path FTIR? There seems to be no benefit in terms of the achievable absorbance resolution, and the measurement precision is comparable. One argument brought by the authors is the "rapid scan rate" and "faster than turbulence-induced intensity variations", and "instrument-specific calibrations" are not needed. In the reviewer's opinion, it would be beneficial if the authors could discuss and summarize the benefits/drawback compared to open path FTIR in one paragraph.

3. The principle of dual comb technology should be briefly stated using mathematical formulas, so that it is easier for readers to follow. For example, formulas can be written below "two combs with nominal repetition rates of $f_r$ and offset by $\Delta f_r$ are phase locked together, transmitted through a sample, and their heterodyne signal measured on a photodetector. The resulting rf frequency comb can be mapped back to the optical domain to generate an overall spectrum".

4. The authors state that "the instrument line shape is the sum of two Delta impulses as shown in Fig. 1 (b)". The instrument line shape is not visible in Fig. 1 (b). It would be beneficial if the ILS can be schematically shown with the spectral spacing between the two impulses indicated.

5. It is difficult to assess the measurement precision of atmospheric open-path instruments because the measurement conditions (P/T/vmr) are usually not stable and constant. The authors use the measurements in 6 hours well-mixed time period to calculate Allan deviations. Can the authors draw any conclusions regarding to the 1000s turning point in the Allan plots? Is it instrument drift or given by variations of the atmospheric conditions?

6. The authors have done Allan analysis to assess the precision of individual DCS. The reviewer thinks that it would be interesting to conduct the Allan analysis for the measured concentration differences (Fig. 6). On one hand, it will give an indication of the precision of the differential measurements, on the other hand, the atmospheric influences will be cancelled out, which could be advantageous for conducting Allan analysis. Differential column measurements have been recently successfully used for determining local/city emissions [1, 2]. In [1] the measured differences between two side-by-side solar-viewing FTIRs are analyzed using Allan analysis, to determine the precision of the differential system that consists of two spectrometers (0.01% for XCO2 and XCH4 over 10-minute integration time) and the precision of an individual instrument (assuming the instruments are the same and the measurement noises are statistically uncorrelated). [1] presents a new way to determine the precision of atmospheric measurements, and could be included in the references.

7. Statistical distribution of the differences (caption of Fig. 7): if the Allan deviation of the differences follows a square root law (inversely proportional to the square root of the integration time), the distribution widths should be the third when increasing the integration time from 32s to 5 min (factor 10).

Specific comments:

1. Line 16: better to write "$5 \cdot 10^{-4}$ in absorbance"

2. Line 17: better to write "path-integrated concentrations for carbon dioxide (CO2)"

3. Line 19: averaging time interval information is missing: at 32 s integration time

4. Line 125: "with nominal repetitions rates $f_r$ and the difference in the repetition rates $\Delta f_r$ "

5. Line 132: absolute frequency accuracy is written in wavenumber, it would be beneficial if it is also converted to frequency, i.e. 1 MHz.

6. Line 220: which parallel surfaces cause these etalons?

7. Fig. 4: big discrepancies around 6290 cm$^{-1}$ in both (c) the differences between absorption spectra and (d) fitting residual, please specify the reason for it.

8. Line 644, caption of Fig. 8: ...highlighted in Fig. 3 -> it should be Fig. 5

References:

[1] Chen, J., Viatte, C., Hedelius, J. K., Jones, T., Franklin, J. E., Parker, H., Gottlieb, E. W., Wennberg, P. O., Dubey, M. K., and Wofsy, S. C.: Differential column measurements using compact solar-tracking spectrometers, Atmos. Chem. Phys., 16, 8479-8498, doi:10.5194/acp-16-8479-2016, 2016.

[2] Hase, F., Frey, M., Blumenstock, T., Groß, J., Kiel, M., Kohlhepp, R., Mengistu Tsidu, G., Schäfer, K., Sha, M. K., and Orphal, J.: Application of portable FTIR spectrometers for detecting greenhouse gas emissions of the major city Berlin, Atmos. Meas. Tech., 8, 3059-3068, doi:10.5194/amt-8-3059-2015, 2015.

---

## Author Comment (AC1) · 28 Jun 2017

We have provided the editor and reviewer comments below in black text with our responses below the comments in blue text. Line numbers in our responses refer to the line numbers in the markup version of the revised text.

Editor Comment:

The uncertainty of the H2O concentration measurements will translate into the uncertainty of CO2 and CH4, which will be significant, however, was not included in the list of systematic uncertainties in Table 2.

The uncertainty in the retrieved water concentration is set by the maximum 10% error in $H_2O$ due to the HITRAN linestrengths. (The typical uncertainty on the $H_2O$ fit is approximately 65 ppm, which is negligible in comparison.) At our ~1% water concentration, this leads to a ~0.1% additional uncertainty in the dry mixing ratios. A row has been added to Table 2 to account for this uncertainty.

Reviewer 1 Comments:

This manuscript, entitled "Intercomparison of Open -Path Trace Gas Measurements with Two Frequency Comb Spectrometers," reports on a quantitative evaluation of atmospheric trace gas measurements based on dual-comb spectroscopy. Thanks to their well-polished dual-comb spectrometers and analytical approach, the retrieved dry mole fractions agree to 0.57 ppm for $CO_2$ and 7 ppb for $CH_4$ between the two measurement systems. These results are excellent, while there are some obscure points in the manuscript. Therefore, I recommend the manuscript for publication if following comments and questions are addressed.

We thank Reviewer 1 for taking the time to review this work and for his or her helpful comments.

[Specific comments]
1. While I am briefly familiar with the technique of gas spectroscopy, I am not an expert in atmospheric measurement and concerned about some technical descriptions.

1) Is it OK for AMT readers to use some technical terms such as "WMO-calibration" and "WMO compatibility goal" without any simple explanation?

We have clarified "WMO-calibration" at lines 72-74 which now read "We also compare the DCS retrievals to a cavity ringdown point sensor located near the path that has been tied to the World Meteorological Scale (WMO) manometric scale through calibration with WMO-traceable gases."

The WMO compatibility standards are listed on line 295 with a reference. We have changed the phrase at line 475 to read "WMO compatibility standards" instead of "WMO compatibility goals" to make the language identical.

2) L207: I was confused with the expressions of concentration. Is it correct that dry and wet concentrations of carbon dioxide are expressed as "$XCO_2$" and "$CO_2$," respectively?

Yes, the reviewer is correct. We have made no changes at now line 248, however footnote 1 has been modified for clarity on this issue, as noted below.

3) L209: "Volume mole fraction" might be simply "mole fraction."
We agree and have removed the word "volume" at now line 250.

4) Figure 5: I found volume percentage is normally expressed as "v/v%." Is "%v/v" OK too?
We have changed "%v/v" to simply "%" throughout the paper because we are referring to mole fraction rather than volume fraction in this work. We have specifically updated Figure 5 and its caption and Figure 9.

5) According to the footnote in P2, "ppm" and "ppb" are used for dry concentration (dry mole fraction) and "%v/v" is for wet concentration (wet mole fraction) as in Figure 5. However, "ppm" is used for ΔHDO and ΔH₂O in Figure 6 and 7.
We have modified the footnote to clarify that we use ppm and ppb to mean micro or nanomoles of gas per mole of air, and that we specifically mean per mole of dry air when we use XCO₂ or XCH₄. The footnote now reads "We use dry mole fraction for carbon dioxide and methane, denoted respectively as XCO2 in units of ppm, which are micromoles of CO2 per mole of dry air, or XCH4 in units of ppb, which are nanomoles of CH4 per mole of dry air."

6) I think the explanation about dual-comb spectroscopy is a little insufficient. For example, 1) L94 and 97: Authors should refer to Figure 1(a) here instead of Figure 1 and 1(b).
We have rewritten this paragraph and modified Figure 1 in an attempt to convey dual-comb spectroscopy more clearly. We have also separated the more general discussion of dual-comb spectroscopy (now section 2.1) from the more specific instrument discussion (now section 2.3) to try to improve the clarity.

The rewritten paragraph at lines 112-120 reads:
    A frequency comb is a laser pulsed at a very precise repetition rate of $f_r$ (Cundiff and Ye, 2003; Hall, 2006; Hänsch, 2006). Because the pulse rate is so precisely controlled, this creates a spectrum consisting of very narrow, evenly-spaced modes called comb teeth. Dual frequency comb spectroscopy combines two of these combs that have very slightly different pulse repetition rates that differ by $\Delta f_r$, sends the light through the sample, and on to a detector (see Fig. 1a) (Schiller, 2002; Schliesser et al., 2005; Coddington et al., 2008, 2016; Ideguchi, 2017). It is also possible to transmit only a single comb through the air to measure both dispersion and absorbance (Giorgetta et al., 2015). The basic technique of dual-comb spectroscopy is illustrated in Figure 1 and described in more detail in the literature (Schiller, 2002; Schliesser et al., 2005; Coddington et al., 2008, 2016).

7) L102: I could not understand the explanation "the instrument lineshape is effectively the sum of two delta-functions." What does it mean?
Since both combs pass through the gas, the teeth of both combs are attenuated. These teeth are "delta functions" in frequency as they are very narrow and at precisely known frequencies. The measured signal is the product of the two attenuated teeth, and therefore probes the sample absorption at the two teeth locations. That is what we meant by the sum of the two delta functions. In the reworded Section 2.3 we have added at lines 177-182 "As shown in Fig. 1a, the effective lineshape for each sampled point is well approximated as two closely separated delta-functions located at the known optical frequencies of the two comb lines that are

heterodyned to produce the measured rf signal (e.g. consider the solid and dashed yellow optical comb teeth that lead to the single solid yellow rf comb tooth.) The separation of the two delta-functions (comb teeth) is negligible compared to the ~5-GHz wide absorption lines but can be exactly incorporated in the spectral model."

8) L139: Readers might not be sure whether $f_r$ is a sampling rate or bandwidth.
We have clarified this to "digitized at a sampling rate $f_r$." at now line 171.

9) L183: Please define $Cn^2$.
We have now defined $Cn^2$ as the refractive index structure parameter in the text at lines 223-234.

10) In Figure 4, the observed HDO is $10^{-4}$ %v/v level, whereas ΔHDO is 1000 ppm level in Figure 6 and 7. Are they consistent?
Thank you. The previous figures included a built-in HITRAN isotope scaling factor that we had forgotten to remove. It has now been removed for the HDO concentration. We have also updated the width of the HDO histogram at lines 292-293.

11) In the caption of Figure 8, "40 ppmv/$\sqrt{\tau}$" "4 ppbv/$\sqrt{\tau}$" might be "40 ppm/$\sqrt{\tau}$/s" and "4 ppb/$\sqrt{\tau}$/s," respectively.
The caption now reads "ppm" and "ppb" rather than "ppmv" and "ppbv" and we added "where τ is in seconds".

[Technical corrections]
1. There are some notations without space between the value and unit; for example "1-10s" in L46, "10%" in L51. Please check and correct them.
At line 47, we have changed the text to read "one to tens of kilometers". We have corrected the "10%" to "10 %"at L52 and numerous other instances of this error and additional inconsistencies with the AMT style guide.
2. Notation variability, "dual comb" and "dual-comb."
We have changed all "dual comb" instances to read "dual-comb" (except for one instance that occurred in the references).
3. L94: Reference (Ideguchi, 2017) is missing in the list of references.
We have added this to the reference list at line 563.
4. L214 -215: Notation variability, "three-hour " and "3-hour."
We have corrected to be consistent.
5. L217: A beginning parenthesis is missing.
We have removed the extra ).

Reviewer 2 Comments:

This paper compares the performance of two open-path dual comb spectroscopy (DCS) instruments over two weeks and analyzes the measurement differences, also between the DCSs and a Cavity ring down spectrometer measuring in-situ. The paper describes the function principle of the dual comb spectrometers, the optical setup, and the data processing. The precision of the individual DCS is determined using Allan analysis.

This kind of inter-comparison has not been done so far, and the DCS instruments deployed here are novel. The descriptions and analyses are clear structured. The reviewer recommends the publication of this paper, if the following items are addressed.

We thank Reviewer 2 for taking the time to review this work and for his or her helpful comments.

General comments:

1) As the authors noted, the dual comb spectrometer has negligible line width (120 kHz) and high spectral resolution, the absorbance resolution is below $10^{-3}$ ($5\times10^{-4}$), which is limited by the instrument noise. However, the laser spectroscopic instrument typically has an absorbance resolution of $10^{-5}$, limited by the shot noise of the photodiode (also depends on the incoming light intensity). Could the authors be more specific about the origin of the noise? What kind of instrumental noises are these, photodetector noise, noise from the dual comb laser?

The comparison emphasizes that the two DCS spectra agree to within their instrument noise. This would generally not be the case between laser spectroscopic instruments unless they had precisely the same frequency axis (otherwise one would effectively observe a derivative of the spectrum with higher amplitude than $1e^{-4}$.)

We have added a brief noise discussion in Section 3.1 at lines 259-270 as follows "The difference of the absorption spectra, shown as the black line in Figure 4(c), has a standard deviation of $9\times10^{-4}$ with no observable structure at absorption lines. This difference is dominated by an etalon on the off-axis telescope used with DCS A. After manually fitting out the etalon structure, the remaining difference between DCS A and DCS B is attributed to measurement noise. DCS A has higher return power (see Fig. 3b) and the measurement noise is primarily from relative-intensity noise (RIN) on the comb light. This RIN is mainly white but has a small peak near 14 MHz, which is mapped to ~6290 $cm^{-1}$ in the optical domain, leading to the observed noise increase in that spectral region. DCS B has lower return power and the measurement noise is from the detector. Nevertheless, the two spectra agree to better than $5\times10^{-4}$ over the full spectral region (with the exception of the 7 $cm^{-1}$ section at 6290 $cm^{-1}$), and better than $2.5\times10^{-4}$ over the region near 6100 $cm^{-1}$ where both DCS systems have significant returned optical power."

2) What benefits in the authors' opinion could be gained from the open-path DCS techniques compared to open path FTIR? There seems to be no benefit in terms of the achievable absorbance resolution, and the measurement precision is comparable. One argument brought by the authors is the "rapid scan rate" and "faster than turbulence-induced intensity variations", and "instrument-specific calibrations" are not needed. In the reviewer's opinion, it would be beneficial if the authors could discuss and summarize the benefits/drawback compared to open path FTIR in one paragraph.

We have followed the reviewer's suggestion and combined the comments on the advantages of DCS into a discussion at the end of the new Section 2.1. We do distinguish between open path FTIR instruments for horizontal column instruments and solar-tracking FTIR instruments that make full vertical column measurements. The open-path horizontal FTIR instruments operate with a resolution of 0.5 to 1 $cm^{-1}$ resolution while the pressure-broadened gas lines are approximately 0.15 $cm^{-1}$ (5 GHz) wide. This leads to the necessity of measuring the instrument lineshape (ILS) and incorporating it into the analysis. On the other hand, DCS measures with a

point spacing and instrument lineshape much smaller than the gas absorption lines. The DCS should be much less susceptible to systematics and can even probe the accuracy of spectral models. Other advantages include the collimated laser beam for long distance operation compared to the sub-km range of open path FTS instruments and the insensitivity to turbulence. However, unlike the careful comparisons of solar-looking FTS, it is very challenging to find careful comparisons of open-path FTS in the literature. Indeed, we find no examples of direct horizontal open-path FTS comparisons for greenhouse gases (again as compared to either vertical solar-looking FTS or the data here).

The new paragraph appears at lines 121-138 with the following text:

A DCS system can be thought of as a high-resolution Fourier-Transform spectrometer but has a number of attributes that distinguish it from a conventional horizontal open-path FTS and other open-path instruments that could lead to higher performance atmospheric trace gas monitoring. A compact, mobile DCS system such as this one has no moving parts but dense point spacing (200 MHz or 0.0067 $cm^{-1}$ in this work), effectively no instrument lineshape, and a calibration-free wavelength axis as described in (Rieker et al., 2014; Truong et al., 2016). As a result, it oversamples the 5 GHz-wide (0.15 $cm^{-1}$) pressure-broadened gas lines of carbon dioxide, methane, water and other small molecules without distortion, which should suppress any instrument-specific systematics and allow comparison of DCS data between instruments and over time. Specifically relevant to open-path measurements, the comb output is a diffraction-limited eye-safe laser beam and can support much longer distances than typical open-path FTS systems; here we demonstrate 2-km round-trip measurements, but we have unpublished data for up to 11.6 km round-trip. Finally, unlike swept laser systems, DCS measures all wavelengths at once rather than sequentially and is therefore much more immune to turbulence effects as described in (Rieker et al., 2014). There are still disadvantages. The current system is not yet turn-key and requires intermittent manual adjustments. The shape of the comb spectrum can vary with wavelength and time, thus requiring a real-time reference to retrieve broad-band molecular absorption lines, and finally the spectral width is narrower than an FTIR. However, none of these disadvantages are fundamental but rather technical challenges to be solved.

3) The principle of dual comb technology should be briefly stated using mathematical formulas, so that it is easier for readers to follow. For example, formulas can be written below "two combs with nominal repetition rates of $f_r$ and offset by $\Delta f_r$ are phase locked together, transmitted through a sample, and their heterodyne signal measured on a photodetector. The resulting rf frequency comb can be mapped back to the optical domain to generate an overall spectrum".

We have modified Figure 1 and its caption significantly to try to better convey dual-comb spectroscopy. The mathematical formulae are not very helpful compared to this picture but we add explicit references to places where the math is given. (The math requires a more detailed discussion of the comb and its locking conditions, which would be a significant detour for this paper.) We have also modified the discussion of dual-comb spectroscopy, as noted in response to reviewer 1, to separate out the more general description in Section 2.1 from the details of Section 2.3.

4) The authors state that "the instrument line shape is the sum of two Delta impulses as shown in Fig. 1 (b)". The instrument line shape is not visible in Fig. 1 (b). It would be beneficial if the ILS can be schematically shown with the spectral spacing between the two impulses indicated.

We have updated Fig. 1(b) and reworded this discussion as noted in response to reviewer #1 point 7 above.

5. It is difficult to assess the measurement precision of atmospheric open-path instruments because the measurement conditions (P/T/vmr) are usually not stable and constant. The authors use the measurements in 6 hours well-mixed time period to calculate Allan deviations. Can the authors draw any conclusions regarding to the 1000s turning point in the Allan plots? Is it instrument drift or given by variations of the atmospheric conditions?

As discussed in the next point, we have taken up the suggestion of the reviewer and added an Allan deviation of the difference between the instruments. This separates out the atmospheric and instrument variability. As discussed below and in the new wording in Section 3.3, based on this analysis most of the flattening is likely due to the instrument (but only because we have selected a very well-mixed 6 hour period.)

The new text appears at lines 339-346 and states:

As in (Chen et al., 2016) it is also useful to plot the Allan deviation of the difference in retrieved concentration between the instruments, e.g. $\Delta XCO_2$ and $\Delta XCH_4$ of Figure 6. This removes the atmospheric variability from the Allan deviation and provides information on the relative stability of the two instruments. As it includes contributions from both DCS instruments, it lies above both individual Allan deviations but similarly reaches a floor at ~1000 s, indicating the floor of the individual Allan deviations is likely dominated by instrument rather than atmospheric variability for these data. We note a similar floor is found in (Truong et al., 2016) for static laboratory cell data where it was attributed to the existence of an etalon.

6. The authors have done Allan analysis to assess the precision of individual DCS. The reviewer thinks that it would be interesting to conduct the Allan analysis for the measured concentration differences (Fig. 6). On one hand, it will give an indication of the precision of the differential measurements, on the other hand, the atmospheric influences will be cancelled out, which could be advantageous for conducting Allan analysis. Differential column measurements have been recently successfully used for determining local/city emissions [1, 2]. In [1] the measured differences between two side-by-side solar-viewing FTIRs are analyzed using Allan analysis, to determine the precision of the differential system that consists of two spectrometers (0.01% for XCO2 and XCH4 over 10-minute integration time) and the precision of an individual instrument (assuming the instruments are the same and the measurement noises are statistically uncorrelated). [1] presents a new way to determine the precision of atmospheric measurements, and could be included in the references.

We agree and have added Allan deviation of the differences of $XCO_2$ and $XCH_4$ to the new Figure 8 along with a reference to [1] of the reviewer comments. As suggested, it is useful to examining both the individual and differential Allan deviations since it clarifies that most of the "flattening" at 1000s is likely due to instrument effects rather than atmospheric variability.

We have left Figure 7 that shows the histogram of the differences because it conveys both the precision at a particular time (via the width) and the absolute difference (via the center position) between the retrieved concentrations. A potential strength of DCS is that there is no instrument-specific calibration and it was important to quantify whether the two instruments retrieved the

same concentration; the Allan deviation does not capture that. In other words, one could measure a very low Allan deviation but still have a very large offset between instruments.

7. Statistical distribution of the differences (caption of Fig. 7): if the Allan deviation of the differences follows a square root law (inversely proportional to the square root of the integration time), the distribution widths should be the third when increasing the integration time from 32s to 5 min (factor 10).
Yes, we agree and the broader width at 5 min reflects the instrument drift at that time scale. As the reviewer suggested in the previous comment, this effective floor with averaging time is captured best in the Allan deviation that has been added to Figure 8. (See answer to comment above.)

Specific comments:
1. Line 16: better to write "$5 \times 10^{-4}$ in absorbance"
We have made this change.
2. Line 17: better to write "path-integrated concentrations for carbon dioxide (CO2)"
We have made this change.
3. Line 19: averaging time interval information is missing: at 32 s integration time
We have made this change (now at line 20).
4. Line 125: "with nominal repetitions rates $f_r$ and the difference in the repetition rates $\Delta f_r$"
We have made this change (now at line 159).
5. Line 132: absolute frequency accuracy is written in wavenumber, it would be beneficial if it is also converted to frequency, i.e. 1 MHz.
We have made this change (now at line 164).
6. Line 220: which parallel surfaces cause these etalons?
It is unclear (or we would remove them). They could be in one of the fiber-optic components or in the free-space optics. We have removed as many etalons as possible in the design but since we do not know the origin of these, we have not speculated.

7. Fig. 4: big discrepancies around 6290 cm$^{-1}$ in both (c) the differences between absorption spectra and (d) fitting residual, please specify the reason for it.
This noise originates from excess laser noise on one of the systems that is related to the use of semiconductor saturable absorber mirror (SESAM) (see Sinclair et al. (2015) Rev. Sci. Inst. for further details). The noise appears at ~14 MHz in the RF domain, but is scaled up to the optical along with the rest of the RF comb, ending up at ~188 THz/6290 cm$^{-1}$.
Section 3.1 now has the added words that explain this noise as discussed in point 1 above.

8. Line 644, caption of Fig. 8: ...highlighted in Fig. 3 -> it should be Fig. 5
Thank you, we have fixed this.

References cited by the authors:
Coddington, I., Newbury, N. and Swann, W.: Dual-comb spectroscopy, Optica, 3(4), 414, doi:10.1364/OPTICA.3.000414, 2016.

Sinclair, L. C., Deschênes, J.-D., Sonderhouse, L., Swann, W. C., Khader, I. H., Baumann, E., Newbury, N. R. and Coddington, I.: Invited Article: A compact optically coherent fiber frequency comb, Rev. Sci. Instrum., 86(8), 081301, doi:10.1063/1.4928163, 2015.

Truong, G.-W., Waxman, E. M., Cossel, K. C., Baumann, E., Klose, A., Giorgetta, F. R., Swann, W. C., Newbury, N. R. and Coddington, I.: Accurate frequency referencing for fieldable dual-comb spectroscopy, Opt. Express, 24(26), 30495, doi:10.1364/OE.24.030495, 2016.